# Daytime napping, sleep duration and serum C reactive protein: a population-based cohort study

Yue Leng,[1] Sara Ahmadi-Abhari,[1] Nick W J Wainwright,[1] Francesco P Cappuccio,[2] Paul G Surtees,[1] Robert Luben,[1] Carol Brayne,[3] Kay-Tee Khaw[1]

[1]Department of Public Health and Primary Care, Strangeways Research Laboratory, University of Cambridge, Cambridge, UK
[2]Division of Mental Health & Wellbeing, Warwick Medical School, University of Warwick, Coventry, UK
[3]Department of Public Health and Primary Care, Institute of Public Health, University of Cambridge, Cambridge, UK

**Correspondence to**
Yue Leng;
yl411@medschl.cam.ac.uk

## ABSTRACT

**Objectives:** To explore whether daytime napping and sleep duration are linked to serum C reactive protein (CRP), a pro-inflammatory marker, in an older aged British population.

**Design:** Cross-sectional study.

**Setting:** European Prospective Investigation into Cancer and Nutrition (EPIC)-Norfolk study.

**Participants:** A total of 5018 men and women aged 48–92 years reported their sleep habits and had serum CRP levels measured.

**Outcome and measures:** CRP was measured (mg/L) during 2006–2011 in fresh blood samples using high-sensitivity methods. Participants reported napping habits during 2002–2004, and reported sleep quantity during 2006–2007. Multivariable linear regression models were used to examine the association between napping and log-transformed CRP, and geometric mean CRP levels were calculated.

**Results:** After adjustment for age and sex, those who reported napping had 10% higher CRP levels compared with those not napping. The association was attenuated but remained borderline significant (β=0.05 (95% CI 0.00 to 0.10)) after further adjustment for social class, education, marital status, body mass index, physical activity, smoking, alcohol intake, self-reported health, pre-existing diseases, systolic blood pressure, hypnotic drug use, depression and in women-only hormone replacement therapy use. The geometric means (95% CI) of CRP levels were 2.38 (2.29 to 2.47) mg/L and 2.26 (2.21 to 2.32) mg/L for those who reported napping and no napping, respectively. A U-shaped association was observed between time spent in bed at night and CRP levels, and nighttime sleep duration was not associated with serum CRP levels. The association between napping and CRP was stronger for older participants, and among extremes of time spent in bed at night.

**Conclusions:** Daytime napping was associated with increased CRP levels in an older aged British population. Further studies are needed to determine whether daytime napping is a cause for systemic inflammation, or if it is a symptom or consequence of underlying health problems.

## INTRODUCTION

Recently there has been growing interest in studying the health implications of daytime napping. Although the benefits of napping

### Strengths and limitations of this study

- Our study benefits from a large population-based sample, measures of daytime as well as nighttime sleep, and a wide range of covariates available for adjustment.
- Sleep habits were self-reported.
- The purpose and duration of napping was not reported, and sleep apnoea was not measured.

on wakefulness performance have been widely documented, especially among shift workers,[1] a few studies also reported an increased health risk associated with daytime napping. Specifically, daytime napping has been associated with increased mortality,[2 3] cardiovascular[4] and diabetes risk.[5] We have recently found a 32% increased mortality risk among those who napped for ≥1 h/day in a middle to older aged British population.[6] While it is not yet clear whether daytime napping is a cause or symptom of the increased health risk, examination of the physiological correlations of napping will help to gain understanding of the napping–health relationship.

One important proposed pathway for the association between sleep and health outcomes is through inflammation.[7] C reactive protein (CRP), a general marker for inflammation, was first suggested to increase after sleep deprivation in an experimental study.[8] Since then, there have been increasing numbers of observational studies which produced mixed findings on the association. Long sleep duration has been associated with increasing CRP among men,[9] women[10] as well as in the sexes combined sample,[11 12] with long sleep being brought up as a potential marker of underlying inflammatory illness.[11] By contrast, Miller et al[13] found an association between short sleep duration and increased CRP only in women. Taheri et al[14] suggested no significant association between

CRP levels and sleep duration in the Wisconsin Sleep Cohort Study.

Despite the growing interest into the inflammatory correlates of nighttime sleep, there has been limited evidence on the physiological effects of daytime napping. Experimental study has reported a beneficial effect of napping on immune cells following acute sleep restriction.[15] However, habitual daytime napping as a lifestyle might relate to different physiological effects compared with napping as a form of recovery sleep, and understanding the link between habitual napping and inflammation could help to clarify the association observed for napping and the increased mortality risk. Moreover, a number of changes in sleep from middle to older adulthood have been reported, including an advanced circadian pacemaker, decreased sleep efficiency and increased daytime sleepiness.[16–18] Therefore we set out to examine the associations between daytime napping and serum CRP levels in the European Prospective Investigation into Cancer (EPIC)-Norfolk cohort, a middle to older aged British population. In addition, we have previously reported the different implications between sleep duration and time spent in bed,[19] so confirmatory analysis was also conducted to test the associations separately for sleep duration and time spent in bed at night. Given the gender disparity suggested by some previous studies,[13 20] we explored sex differences in the associations.

## METHODS
### Participants
The design and study methods of EPIC-Norfolk have been described previously.[21] Briefly, 25 639 men and women were recruited into the EPIC-Norfolk study during 1993–1997 using general practice age–sex registers, and attended the baseline health check. These participants were then followed up for two further health examinations from 1996 to 2000 and from 2006 to 2011. In between these health examinations, participants were sent questionnaires for completion and returned by post. The Norwich District Ethics Committee approved the study and all participants gave signed informed consent.

### Sleep measures
Habitual daytime napping was ascertained during 2002–2004, by asking participants the question "Do you normally take a nap during the day?"

'Nighttime sleep duration' and 'Time spent in bed at night' were ascertained during 2006–2007, by asking participants "On average, about how many hours have you slept each night?"; "At what time do you normally get up?" and "At what time do you normally go to bed?". 'Time spent in bed' was derived from the differences between rise time and bedtime, and as a weighted mean measure of weekday and weekend times (5/7×(time on a weekday)+2/7×(time on a weekend day)).

### C reactive protein
CRP was measured (mg/L) at the third health examination (2006–2011) in fresh blood samples using high-sensitivity methods using the Siemens Dimension clinical chemistry analyser (Newark, Delaware, USA), with between-batch coefficient of variation (CV) values of 3.1% at 7.86 mg/L and 3.7% at 88.75 mg/L.

### Covariates
Covariates measured closer in time to each measure of sleep were chosen and included in the models for the corresponding sleep measures. The covariates were chosen as a priori based on their link with CRP and sleep.[19 22] Social class (professionals, managerial and technical occupations, skilled workers subdivided into non-manual and manual, partly skilled workers and unskilled manual workers) and education (highest qualification attained: no qualifications, educated to age 16, educated to age 18 and educated to degree level) were assessed at the baseline questionnaire. Body mass index (BMI; weight in kilograms divided by height in metres squared) and systolic blood pressure (SBP; mm Hg) were objectively assessed through the health examination. The other covariates reported by questionnaires included: marital status (single, married, widowed, separated and divorced), physical activity (inactive, moderately inactive, moderately active and active[23]), smoking status (current, former and non-smokers), alcohol intake (units of alcohol drunk per week), self-reported general health (excellent, good, moderate or poor), major depressive disorder (MDD; yes/no), pre-existing diseases (including self-reported stroke, myocardial infarction, diabetes, cancer, asthma, bronchitis and emphysema), use of hypnotic drugs (yes/no) and hormone replacement therapy (HRT) use.

### Statistical analysis
Analysis was restricted to those participants with complete data on all covariates. A total of 5018 participants had complete data on CRP and at least one of the sleep measures, and were included in the analysis.

The CRP values were natural-log transformed ($\log_e$ CRP) to approximate a normal distribution. Owing to the U-shaped relationship reported by previous studies on sleep duration and health risk,[24 25] and in order to retain sufficient numbers, sleep duration was categorised into three categories (<6, 6–8 and >8 h) and time spent in bed was categorised into four categories (<6, 6–8, 8–10 and >10 h). Multivariable linear regression models using $\log_e$ CRP as the outcome were conducted to examine the associations between each sleep measure (daytime napping (N=4712), time spent in bed (N=4476) and sleep duration (N=4795)) and CRP levels. For each sleep variable, results were presented: (A) adjusted for age and sex; (B) further adjusted for social class, education and marital status, BMI, physical activity, smoking and alcohol intake, depression, self-reported health, pre-existing diseases, SBP, hypnotic drug use and in women-only postmenopausal HRT. The sleep

measures were mutually adjusted in model B. Since the associations for sleep duration and time spent in bed might be non-linear (approximately U-shaped), the β coefficients (with 95% CIs) were calculated with 6–8 h being the reference groups for the examination of sleep quantitative measures. For analysis by napping habit, the no napping category was chosen as reference group. Results were presented for the whole sample and for men and women separately.

Geometric—mean (95% CI) of serum CRP values (mg/L) were calculated by exponentiating crude and adjusted-least square mean (95% CI) $\log_e$ CRP values. Finally, subgroup analysis was conducted for the main sleep variables of interest, napping and time spent in bed at night, to explore the effects of potential effect modifiers (including age, sex, social class, smoking, BMI, physical activity and pre-existing diseases). These analyses were adjusted for age, sex, social class, education, marital status and BMI. The two-by-two interactions between napping habit, time spent in bed and sleep duration were examined using the likelihood ratio test. Sensitivity analysis: all statistical tests were two-sided, and a p value <0.05 was considered statistically significant. Analyses were implemented in STATA V.12.0.

## RESULTS

Characteristics of the study sample (2267 men and 2751 women) are shown in table 1. The mean age of the participants was 69.5 (ranging from 48 to 92) years old. More than 60% of the sample came from non-manual social class, and 48% were former or current smokers. A total of 38.5% men and 23.1% women reported daytime napping. The average time people spent in bed every night was 8.6±0.8 h and the average sleep duration was 6.9±1.1 h. Serum CRP ranged from 0.1 to 116.6 mg/L, with a median of 2.0 mg/L (IQR 2.07 mg/L).

The unadjusted geometric means of CRP were 2.53 (2.44 to 2.63) and 2.20 (2.15 to 2.26) mg/L for those who reported napping and no napping, respectively. Table 2 shows differences in $\log_e$ CRP according to the report of napping. After adjustment for age and sex, daytime napping was associated with 10% higher CRP levels (β=0.10, 95% CI (0.05 to 0.15), p<0.001). With further adjustment for socioeconomic status, health-related behaviours, physical health and nighttime sleep, the association was attenuated but remained borderline significant (β=0.05, 95% CI (0.00 to 0.10), p=0.048). Table 3 shows the multivariable-adjusted geometric means of CRP. The geometric means of CRP levels were 2.38 (2.29 to 2.47) and 2.26 (2.21 to 2.32) mg/L for those who reported napping and no napping, respectively. When analyses were stratified by sex, the associations between the above napping measure and CRP levels were only significant among women.

The unadjusted geometric means of CRP for those who reported spending <6, 6–8, 8–10 and >10 h in bed at night were 3.70 (2.48 to 5.53), 2.25 (2.15 to 2.35),

2.29 (2.23 to 2.35) and 2.84 (2.52 to 3.19) mg/L, respectively. After adjustment for age and sex, spending <6 and >10 h in bed were associated with 51% and 17% increase in CRP levels, respectively (table 2). After full adjustments, long time spent in bed was associated with 12% increase in CRP levels while the association between short time spent in bed and $\log_e$ CRP was not significant. The geometric means of CRP levels among those who spent >10 and <6 h in bed were 2.57 (2.30 to 2.87) and 3.00 (2.04 to 4.40) mg/L, respectively, higher than those who spent 6–8 h in bed (2.28 (2.19 to 2.38) mg/L). Nighttime sleep duration was not associated with $\log_e$ CRP. Repeating the analyses restricted to individuals with CRP levels below 10 mg/L did not change the results.

Subgroup analysis investigated differences in these associations by age, sex, social class, smoking, BMI, physical activity and pre-existing diseases (table 4). These data suggested that the association between napping and CRP levels was stronger for those who were older as compared with those who were younger (p for interaction=0.007). In addition, the association was more pronounced for extremes of time spent in bed at night (p for interaction=0.01; figure 1).

## DISCUSSION

Our findings from more than 4000 middle to older aged English adults suggest that daytime napping was independently associated with higher CRP levels. Those who reported napping had 10% higher CRP levels after adjustment for age and sex. The association was attenuated but remained statistically significant after adjustment for all the covariates. A U-shaped association was observed between time spent in bed at night and CRP levels, though only the effect of long time spent in bed was statistically significant. The napping–CRP associations were more pronounced among extremes of time spent in bed at night. No association was found between sleep duration and CRP levels.

To the best of our knowledge, this is the first study to report an association between habitual daytime napping and inflammation. This gives biological insights to the health impact of daytime napping proposed by previous studies.[3 26] Our study benefits from a large population-based sample, and a wide range of covariates available for adjustment. Although only moderate effect size was detected in the multivariable model, the possibility of over adjustment cannot be excluded, given the complexity of the relationship between sleep and health. Overall, this preliminary finding at population level provides an interesting perspective and implies the need for further physiological studies. There are several limitations. Similar to most previous population studies,[26 27] daytime napping was self-reported as yes or no. It was suggested that power naps (naps <30 min) can be largely beneficial,[1 28] while naps >30 min can cause sleep inertia and are not recommended.[29] Without detailed report of

**Table 1** Baseline characteristics of the EPIC-Norfolk participants

|  | Men | Women | Total |
|---|---|---|---|
| Age (years) | 70.5 (8.0) | 68.6 (7.9) | 69.5 (8.0) |
| Social class |  |  |  |
| Non-manual | 1495 (66.5%) | 1885 (69.3%) | 3380 (68.0%) |
| Manual | 753 (33.5%) | 836 (30.7%) | 1589 (32.0%) |
| Education |  |  |  |
| Lower | 686 (30.3%) | 1148 (41.7%) | 1834 (36.6%) |
| Higher | 1580 (69.7%) | 1602 (58.3%) | 3182 (63.4%) |
| Marital status |  |  |  |
| Single | 60 (2.8%) | 99 (3.8%) | 159 (3.3%) |
| Married | 1929 (90.4%) | 2017 (77.6%) | 3946 (83.4%) |
| Others | 144 (6.8%) | 484 (18.6%) | 628 (13.3%) |
| Smoking |  |  |  |
| Current smoker | 147 (6.5%) | 194 (7.1%) | 341 (6.8%) |
| Former smoker | 1178 (52.1%) | 881 (32.1%) | 2059 (41.2%) |
| Never smoked | 936 (41.4%) | 1666 (60.8%) | 2602 (52.0%) |
| Physical activity |  |  |  |
| Inactive | 624 (29.0%) | 677 (25.9%) | 1301 (27.4%) |
| Moderately inactive | 534 (24.9%) | 877 (33.6%) | 1411 (29.6%) |
| Moderately active | 497 (23.1%) | 613 (23.5%) | 1110 (23.3%) |
| Active | 493 (23.0%) | 447 (17.1%) | 940 (19.7%) |
| BMI (kg/m$^2$) | 26.7 (3.1) | 26.0 (4.0) | 26.3 (3.6) |
| SBP (mm Hg) | 134.9 (16.9) | 129.9 (17.3) | 132.2 (17.3) |
| Alcohol (units) | 8.0 [3.0–15.0] | 3.0 [1.5–8.0] | 5.0 [2.0–11.0] |
| Pre-existing diseases |  |  |  |
| No | 2095 (97.5%) | 2499 (98.9%) | 4594 (98.2%) |
| Yes | 53 (2.5%) | 29 (1.1%) | 82 (1.8%) |
| Hypnotic use |  |  |  |
| No | 2236 (98.6%) | 2717 (98.8%) | 4953 (98.7%) |
| Yes | 31 (1.4%) | 34 (1.2%) | 65 (1.3%) |
| MDD in the last year |  |  |  |
| No | 1964 (97.1%) | 2336 (94.3%) | 4300 (95.6%) |
| Yes | 58 (2.9%) | 142 (5.7%) | 200 (4.4%) |
| Self-reported general health |  |  |  |
| Excellent | 443 (19.7%) | 564 (20.6%) | 1007 (20.2%) |
| Good | 1549 (68.8%) | 1828 (66.8%) | 3377 (67.7%) |
| Moderate | 245 (10.9%) | 332 (12.1%) | 577 (11.6%) |
| Poor | 15 (0.7%) | 11 (0.4%) | 26 (0.5%) |
| Napping |  |  |  |
| No | 1309 (61.5%) | 1986 (76.9%) | 3295 (69.9%) |
| Yes | 819 (38.5%) | 598 (23.1%) | 1417 (30.1%) |
| Time spent in bed (h) | 8.5 (0.9) | 8.7 (0.8) | 8.6 (0.8) |
| Sleep duration (h) | 7.0 (1.1) | 6.9 (1.1) | 6.9 (1.1) |
| CRP (mg/L) | 1.9 [1.3–3.2] | 2.0 [1.3–3.5] | 2.0 [1.3–3.4] |

Results were presented as mean (SD), median [IQR] and the rest N (%).
BMI, body mass index; CRP, C reactive protein; MDD, major depressive disorder; SBP, systolic blood pressure.

napping duration, we were unable to differentiate the effects of power naps from excessive napping. Other sleep disorders (eg, sleep apnoea) that might have led to daytime sleepiness were not assessed, so the possibility of residual confounding by these sleep disorders cannot be ruled out. Moreover, reported sleep may be a marker of distress levels,[30] which has been linked with CRP levels.[31] However, evaluation of sleep in the primary care setting relies on self-reported data from patients, and also the association remained after adjustment for MDD. Another limitation of this study is that associations are based on a single measurement of CRP. However, the intraindividual variability of CRP measurements is similar to that of other physiological measures such as blood pressure and cholesterol levels, and a single measurement of CRP has been shown to predict a range of health outcomes.[32 33] Besides, previous study has reported no diurnal variation of CRP concentrations.[34] Notably, while the assessment of napping preceded the CRP measures, baseline CRP level was not measured and there is no information on changes in CRP over time. We were therefore unable to distinguish between

**Table 2** Linear regression on sleep and log$_e$ CRP in the EPIC-Norfolk cohort study

| | All (β (95% CI)) | | Men | | | Women | | |
|---|---|---|---|---|---|---|---|---|
| | Model A | Model B | | Model A | Model B | | Model A | Model B |
| Napping (4712) | | | 2128 | | | 2584 | | |
| No (n=3295) | Reference | | 1309 | Reference | | 1986 | Reference | |
| Yes (n=1417) | 0.10*** (0.05 to 0.15) | 0.05* (0.00 to 0.10) | 819 | 0.08* (0.01 to 0.15) | 0.03 (−0.03 to 0.10) | 598 | 0.12*** (0.06 to 0.19) | 0.07 (0.00 to 0.13) |
| Time spent in bed (4476) | | | 2036 | | | 2440 | | |
| <6 h (n=13) | 0.51* (0.11 to 0.91) | 0.27 (−0.11 to 0.66) | 7 | 0.26 (−0.30 to 0.82) | 0.18 (−0.37 to 0.73) | 6 | 0.79** (0.22 to 1.37) | 0.43 (−0.12 to 0.97) |
| 6–8 h (n=1101) | Reference | | 620 | Reference | | 481 | Reference | |
| 8–10 h (n=3208) | −0.01 (−0.06 to 0.04) | 0 (−0.05 to 0.05) | 1342 | −0.03 (−0.10 to 0.04) | −0.01 (−0.09 to 0.06) | 1866 | 0 (−0.07 to 0.07) | 0.02 (−0.05 to 0.09) |
| >10 h (n=154) | 0.17** (0.05 to 0.29) | 0.12* (0.00 to 0.24) | 67 | 0.18 (−0.02 to 0.37) | 0.14 (−0.05 to 0.33) | 87 | 0.17* (0.00 to 0.33) | 0.11 (−0.05 to 0.26) |
| p Value | 0.001 | 0.09 | | 0.11 | 0.35 | | 0.01 | 0.23 |
| Sleep duration (4795) | | | 2173 | | | 2622 | | |
| <6 h (n=1232) | 0 (−0.04 to 0.05) | −0.02 (−0.07 to 0.02) | 476 | −0.01 (−0.09 to 0.06) | −0.05 (−0.13 to 0.02) | 756 | 0.02 (−0.04 to 0.08) | −0.01 (−0.07 to 0.05) |
| 6–8 h (n=3154) | Reference | | 1483 | Reference | | 1671 | Reference | |
| >8 h (n=409) | 0.04 (−0.03 to 0.12) | 0 (−0.07 to 0.08) | 214 | 0.08 (−0.03 to 0.19) | 0.02 (−0.08 to 0.13) | 195 | −0.01 (−0.11 to 0.10) | −0.01 (−0.11 to 0.09) |
| p Value | 0.57 | 0.59 | | 0.28 | 0.31 | | 0.86 | 0.95 |

Model A: age, sex; model B: age, sex, social class, education, marital status, BMI, physical activity, smoking, alcohol, self-reported health, pre-existing diseases, SBP, hypnotic use, major depressive disorder and in women-only HRT use. The sleep measures were mutually adjusted in model B. Results were presented as β coefficients (95% CI) of linear regression models using log$_e$ CRP as the outcome.
*p<0.05; **p<0.01; ***p<0.001.
BMI, body mass index; CRP, C reactive protein; EPIC, European Prospective Investigation into Cancer; HRT, hormone replacement therapy; SBP, systolic blood pressure.

napping as a risk factor for, or as an early marker of, increased inflammatory levels. Finally, the current analysis was restricted to the 5018 participants with complete data on CRP and at least one of the sleep measures. Compared with the other participants from the baseline cohort (N=25 639), the present sample were 4 years younger, more likely to be of higher social class and higher education levels. Therefore, our findings have limited generalisability to populations with lower socioeconomic status.

The present study suggested that daytime napping and long time spent in bed at night were associated with higher CRP levels. The observed effect size was similar to previous report on sleep duration and CRP levels,[10] and the increase in CRP levels associated with napping was similar to that associated with 5-year increase in age.[22] While there are no other studies with which to directly compare our findings, the current study is in line with existing evidence on the increased cardiovascular, metabolic and mortality risks among those who take naps, found both by other studies and in our study population.[5 6 35 36] Meanwhile, a Greek cohort study suggested that siesta might help to reduce coronary mortality through a stress-releasing mechanism in healthy working men.[37] We also found the association between daytime napping and increased CRP levels only significant among participants over 70 years old when analysis was stratified by age. Daytime napping might have different implications for health among different age groups, and this needs to be considered by future studies. Notably, daytime napping or daytime sleepiness has been associated with the occurrence of degenerative diseases such as Parkinson's disease and dementia in the older population.[38 39] It has also been reported that chronic inflammation, as indicated by an elevation in plasma interleukin 6 levels, was predictive of future Parkinson's disease.[40] In the current study, daytime napping was associated with a higher serum CRP level, while information on other inflammatory biomarkers was not obtained. Future studies with more complete measures of chronic inflammation might help to clarify the link between daytime napping and degenerative diseases in the elderly. Several studies suggested that the association between sleep and inflammation was only observed in women.[13 20] Consistent with these findings, the association between napping and increased CRP levels was only significant in women, although test for interaction was not statistically significant. The reason for this is unclear, and the gender disparities of sleep physiology need further exploration.

Time spent in bed was rarely studied in relation to CRP concentrations, but long sleep duration has been associated with increased CRP levels.[9 11] Since epidemiology studies have used mixed ways to define sleep duration, the associations between long sleep and inflammation found by some previous studies might actually reflect the effects of long time spent in bed, which was associated with poor general health.[19] Indeed,

**Table 3** Geometric means (95% CI) of CRP (mg/L) by sleep in the EPIC-Norfolk cohort study

| | All | | Men | | | Women | | |
|---|---|---|---|---|---|---|---|---|
| | Model A | Model B | Men | Model A | Model B | Women | Model A | Model B |
| Napping (4712) | | | 2128 | | | 2584 | | |
| No (n=3295) | 2.23 (2.17 to 2.28)*** | 2.26 (2.21 to 2.32)* | 1309 | 2.18 (2.09 to 2.27)* | 2.22 (2.13 to 2.31) | 1986 | 2.27 (2.20 to 2.35)*** | 2.30 (2.23 to 2.37)* |
| Yes (n=1417) | 2.47 (2.37 to 2.57) | 2.38 (2.29 to 2.47) | 819 | 2.36 (2.24 to 2.49) | 2.29 (2.18 to 2.41) | 598 | 2.57 (2.43 to 2.73) | 2.45 (2.31 to 2.59) |
| Time spent in bed (4476) | | | 2036 | | | 2440 | | |
| <6 h (n=13) | 3.82 (2.56 to 5.69)* | 3.00 (2.04 to 4.40) | 7 | 2.97 (1.70 to 5.21) | 2.73 (1.58 to 4.72) | 6 | 5.09 (2.87 to 9.01)** | 3.46 (2.01 to 5.93) |
| 6–8 h (n=1101) | 2.30 (2.20 to 2.40) | 2.28 (2.18 to 2.38) | 620 | 2.29 (2.16 to 2.43) | 2.27 (2.14 to 2.41) | 481 | 2.30 (2.16 to 2.45) | 2.28 (2.14 to 2.42) |
| 8–10 h (n=3208) | 2.27 (2.21 to 2.33) | 2.29 (2.23 to 2.34) | 1342 | 2.22 (2.14 to 2.32) | 2.24 (2.15 to 2.33) | 1866 | 2.31 (2.23 to 2.38) | 2.32 (2.25 to 2.39) |
| >10 h (n=154) | 2.73 (2.43 to 3.06)** | 2.57 (2.30 to 2.87)* | 67 | 2.74 (2.28 to 3.28) | 2.61 (2.19 to 3.11) | 87 | 2.72 (2.34 to 3.17)* | 2.50 (2.17 to 2.89) |
| Sleep duration (4795) | | | 2173 | | | 2622 | | |
| <6 h (n=1232) | 2.28 (2.18 to 2.37) | 2.24 (2.15 to 2.33) | 476 | 2.19 (2.05 to 2.35) | 2.14 (2.00 to 2.28) | 756 | 2.34 (2.23 to 2.46) | 2.30 (2.19 to 2.41) |
| 6–8 h (n=3154) | 2.27 (2.21 to 2.33) | 2.29 (2.24 to 2.35) | 1483 | 2.22 (2.14 to 2.31) | 2.26 (2.17 to 2.34) | 1671 | 2.30 (2.23 to 2.38) | 2.32 (2.25 to 2.40) |
| >8 h (n=409) | 2.36 (2.20 to 2.53) | 2.30 (2.15 to 2.46) | 214 | 2.41 (2.18 to 2.66) | 2.30 (2.09 to 2.54) | 195 | 2.29 (2.07 to 2.53) | 2.29 (2.09 to 2.52) |

Model A: age, sex; model B: age, sex, social class, education, marital status, BMI, physical activity, smoking, alcohol, self-reported health, pre-existing diseases, SBP, hypnotic use, major depressive disorder and in women-only HRT use. The sleep measures were mutually adjusted in model B. Results were presented as geometric means (95% CI) of CRP (mg/L).
*p<0.05 **p<0.01 ***p<0.001; p values were based on comparisons using no napping or 6–8 h as the reference category.
BMI, body mass index; CRP, C reactive protein; EPIC, European Prospective Investigation into Cancer; HRT, hormone replacement therapy; SBP, systolic blood pressure.

long time spent in bed has been associated with poorer general health, pre-existing health problems and more frequent use of sleep medications in the EPIC-Norfolk cohort.[19] Therefore, the observed association between long time spent in bed and increased CRP levels might be explained by declining health status. We found no association between sleep duration and CRP levels. This is consistent with findings from the Wisconsin Sleep Cohort Study, which showed no association between CRP and sleep duration, measured objectively as well as subjectively.[14] Meanwhile, previous studies have found short as well as long sleep duration to be associated with increased CRP levels.[13] [14] Recently, a longitudinal study suggested that each hour per night decrease in sleep duration was associated with 8.1% higher average levels of CRP over a 5-year period.[41] The different correlations with serum CRP levels between the two sleep quantity measures need to be considered by future studies. Interestingly, the association between napping and CRP levels was more pronounced among extremes of time spent in bed. It is possible that extremes of time spent in bed at night partly reflected disturbed nighttime sleep, and thereby added to the association for daytime napping. While the complex inter-relationship between nighttime and daytime sleep is yet to be confirmed, the present study suggested a need for a thorough consideration of nighttime and daytime sleep in future studies.

It has been suggested that sleep deprivation is related to the activation of the autonomic nervous system and increased catecholamines, and subsequently stimulate the release of inflammatory mediators.[42] It is unclear biologically why daytime napping was associated with increased levels of inflammatory markers. One possible explanation is through a rise in blood pressure or heart rate on awakening from a nap, which might be followed by increased endothelial shear stress and the production of inflammatory mediators.[43] Alternatively, daytime napping could be the consequence of unnoticed nighttime sleep disturbance (eg, sleep apnoea) or sleep deprivation, which have been linked with systemic inflammation.[42] [44] [45] Sleep apnoea was not measured in the current study, but the association between daytime napping and serum CRP levels remained after adjustment for BMI and SBP, two strong correlates of sleep apnoea.

In the UK, daytime napping is not part of the cultural norm. It is plausible that those who reported daytime napping and long time spent in bed may represent a population with underlying ill health, and inflammation could be an early sign of these health problems. Although the association between daytime napping and CRP concentrations remained after adjustment for pre-existing diseases, we did find it more pronounced among the older participants, and there might be more existing health problems among this population. Further studies are needed to determine whether daytime napping is a cause for systemic inflammation, or if it is a symptom or consequence of disturbed sleep at night or underlying health problems.

**Table 4** The association between $\log_e$ CRP and napping or time spent in bed at night, by subgroups

| | Napping (reference: no) Yes | | Time spent in bed at night (reference: 6–8 h) 0–6 h | | 8–10 h | | >10 h | |
|---|---|---|---|---|---|---|---|---|
| | β | (95% CI) | β | (95% CI) | β | (95% CI) | β | (95% CI) |
| Age | p for interaction=0.007 | | p for interaction=0.66 | | | | | |
| <70 | 0 | (−0.06 to 0.06) | 0.14 | (−0.25 to 0.52) | 0 | (−0.06 to 0.06) | 0.16* | (0.01 to 0.32) |
| >70 | 0.09** | (0.02 to 0.15) | 0.02 | (−0.59 to 0.63) | 0.03 | (−0.05 to 0.11) | 0.12 | (−0.04 to 0.28) |
| Sex | p for interaction=0.87 | | p for interaction=0.82 | | | | | |
| Men | 0.04 | (−0.02 to 0.11) | 0.07 | (−0.40 to 0.53) | −0 | (−0.08 to 0.06) | 0.16 | (−0.03 to 0.35) |
| Women | 0.05 | (−0.01 to 0.11) | 0.15 | (−0.33 to 0.62) | 0.02 | (−0.04 to 0.08) | 0.1 | (−0.04 to 0.23) |
| Social class | p for interaction=0.38 | | p for interaction=0.31 | | | | | |
| Higher | 0.03 | (−0.03 to 0.08) | 0.15 | (−0.27 to 0.57) | 0.01 | (−0.05 to 0.06) | 0.19** | (0.05 to 0.33) |
| Lower | 0.07 | (−0.01 to 0.14) | 0.08 | (−0.45 to 0.61) | 0 | (−0.08 to 0.09) | 0 | (−0.18 to 0.19) |
| Smoking | p for interaction=0.5 | | p for interaction=0.55 | | | | | |
| Current or former smokers | 0.01 | (−0.05 to 0.08) | 0.12 | (−0.33 to 0.57) | −0 | (−0.10 to 0.05) | 0.12 | (−0.04 to 0.29) |
| Never smoked | 0.06* | (0.01 to 0.12) | 0.11 | (−0.37 to 0.60) | 0.03 | (−0.03 to 0.09) | 0.11 | (−0.04 to 0.26) |
| BMI | p for interaction=0.73 | | p for interaction=0.30 | | | | | |
| Lower | 0.06 | (−0.00 to 0.12) | −0.15 | (−0.76 to 0.46) | 0.04 | (−0.03 to 0.10) | 0.18* | (0.02 to 0.34) |
| Higher | 0.07* | (0.01 to 0.14) | 0.36 | (−0.05 to 0.77) | −0 | (−0.10 to 0.04) | 0.09 | (−0.07 to 0.25) |
| Physical activity | p for interaction=0.48 | | p for interaction=0.41 | | | | | |
| Inactive | 0.06* | (0.00 to 0.12) | 0.28 | (−0.14 to 0.71) | 0.02 | (−0.04 to 0.08) | 0.13 | (−0.00 to 0.26) |
| Active | 0.02 | (−0.05 to 0.09) | −0.13 | (−0.65 to 0.38) | −0 | (−0.10 to 0.05) | 0.08 | (−0.14 to 0.29) |
| Pre-existing diseases | p for interaction=0.05 | | p for interaction=0.08 | | | | | |
| Yes | 0.05 | (−0.25 to 0.35) | NA | | 0.19 | (−0.19 to 0.56) | 0.81** | (0.21 to 1.41) |
| No | 0.05* | (0.01 to 0.10) | 0.11 | (−0.23 to 0.45) | 0.02 | (−0.03 to 0.07) | 0.08 | (−0.04 to 0.20) |

The model adjusted for age, sex, social class, education, marital status and BMI.
Results were presented as β coefficients of linear regression models using $\log_e$ CRP as the outcome.
*p<0.05 ** p<0.01 ***p<0.001.
BMI, body mass index; CRP, C reactive protein.

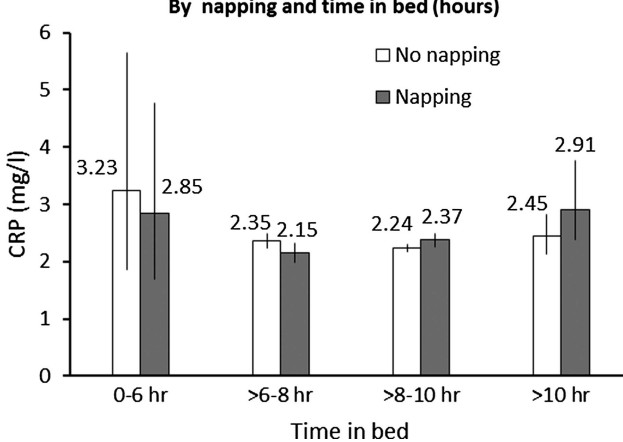

**Figure 1** Serum levels of C reactive protein (CRP) by napping and time spent in bed. Values are geometric mean CRP (mg/L) across categories adjusted for age, body mass index, physical activity, smoking, alcohol intake, social class, education, marital status, major depressive disorder, self-reported health, pre-existing diseases, systolic blood pressure, hypnotic drug use, and in women-only, postmenopausal hormone replacement therapy. The sleep measures were mutually adjusted. Vertical bars represent 95% CIs.

## CONCLUSIONS

In summary, we found for the first time that daytime napping was associated with increased CRP levels in a population-based middle to older aged English cohort. The association between daytime napping and CRP levels was more pronounced among older people, and among those who spent extremes of time in bed at night. While the present study provides new insights into the biological effects of daytime napping, causality cannot be established from an observational study. Daytime napping might be useful as an independent indicator of people who are at underlying health risk or who suffer from disturbed sleep at night. Future physiological studies are needed to gain better understanding of the association.

**Acknowledgements** The authors thank all study participants, general practitioners and the EPIC-Norfolk study team for their contribution.

**Contributors** YL and SA-A analysed the data and wrote the manuscript with coauthors. YL, SA-A, NWJW, FPC, PGS, CB and K-TK discussed the analysis, interpretation and presentation of these data. RL performed all data management and record linkage. RL and K-TK are on the management team of EPIC-Norfolk population study and contributed substantially to acquisition of data. K-TK is a principal investigator in the EPIC-Norfolk study. All authors provided detailed comments on the draft, revised the manuscript critically, and read and approved the final manuscript.

**Funding** The design and conduct of the EPIC-Norfolk study and collection and management of the data was supported by programme grants from the Medical Research Council UK (G9502233, G0401527) and Cancer Research UK (C864/A8257, C864/A2883). YL is supported by the Cambridge Overseas Trust. FPC leads the Sleep Health & Society Programme at the University of Warwick supported, in part, by the University of Warwick RDF and IAS. It has received funding by the NHS National Workforce Projects and the Economic & Social Research Council (ES/K002910/1).

**Competing interests** None.

**Ethics approval** The Norwich District Ethics Committee.

**Provenance and peer review** Not commissioned; externally peer reviewed.

**Data sharing statement** No additional data are available.

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
