## [Reviewer comments · BMJ Open]

This paper was submitted to the JECH but declined for publication following peer review. The authors addressed the reviewers' comments and submitted the revised paper to BMJ Open. The paper was subsequently accepted for publication at BMJ Open.

ARTICLE DETAILS

TITLE (PROVISIONAL)	Daytime napping, sleep duration and serum C-reactive protein: a population-based cohort study
AUTHORS	Leng, Yue; Ahmadi-Abhari, Sara; Wainwright, Nicholas; Cappuccio, Francesco; Surtees, Paul; Luben, Robert; Brayne, Carol; Khaw, Kay-Tee

VERSION 1 - REVIEW

REVIEWER	Honglei Chen, Senior Investigator National Institute of Environmental Health Sciences, USA
REVIEW RETURNED	30-Jul-2014

GENERAL COMMENTS	Leng et al. evaluated associations between daytime napping / sleep duration and level of serum CRP among 5018 participants of the EPIC Norfolk study. The analysis is largely cross-sectional although the assessment for napping was a few years prior to CRP measurement. The authors found that day napping was associated with higher level of CRP while the association between nighttime sleeping/time-in-bed and CRP was u-shaped. This is the first study on napping and CRP, and its findings may help understand some of the previous epidemiological observations on daytime napping and adverse health outcomes among the elderly. Overall the analyses were well performed, particularly the joint analysis on daytime napping and nighttime sleeping. The reviewer however has several comments for authors to consider. 1) Abstract: It is somewhat misleading to call this a prospective study given the short time interval between exposure and outcome and the one-time measurement of outcome.2) Given the average age of the cohort and the stronger findings among elderly, the discussion may be strengthened by commenting on previous findings on napping/sleeping and degenerative diseases, such as dementia and Parkinson's disease (e.g. Gao et al. AJE 2011).3) Given the complex relationship between napping/sleeping and various outcomes, it is not a surprise to see attenuation in the multivariate model which might actually have resulted in some over adjustment; the authors may want to acknowledge this possibility in the discussion.4) Were any other pro-inflammatory markers measured in addition to CRP? Any QC for CRP measurement, if so, please provide the data.5) Table 3 footnote: no explanation for "*..."
---

	6) Table 4: there was a borderline statistical interaction between napping and preexisting diseases, but the Betas from these two groups were identical, please make sure numbers are accurate.
--	---

REVIEWER	Teresa Ward University of Washington USA
REVIEW RETURNED	20-Aug-2014

GENERAL COMMENTS	Comments: The study evaluates daytime napping, sleep duration, and serum C-reactive protein in 5,018 adults 48 to 92 years. Daytime napping, sleep duration, and time in bed (TIB) were obtained via self-report and CRP was measured one time via serum. Covariates –age, sex, BMI, physical activity, smoking, alcohol intake, pre-existing diseases and conditions (systolic BP, depression), and demographics were controlled for in the analysis. Compared to non-nappers, those who napped during the day had increased CRP levels, however these were not statistically different. This finding was more pronounced in older adults, women, and increased TIB. Sleep duration was not a significant predictor of CRP. These findings may reflect the fact that CRP was collected one time after napping data was obtained, higher proportion of women in the sample, and/or 62% of your sample reported no napping. A few comments:  1. Page 6, line 12—“Habitual daytime napping as a lifestyle etc” authors mention different physiological effects compared to napping as a form of recovery sleep. Given the broad age range in your sample (48 to 92 years), further discussion about the changes in sleep from middle adulthood and older adulthood would strengthen this manuscript. Elderly often experience advanced sleep phase and daytime naps are common. 2. Long TIB was associated with increased CRP, and short TIB was not. Further discussion on this finding would strengthen the discussion. amplitude as an arousal marker, wavelet analysis of EEG, etc While power analysis is better than standard stages, it is a limited approach.
--

VERSION 1 – AUTHOR RESPONSE

Reviewer: 1

Reviewer Name Honglei Chen, Senior Investigator

Leng et al. evaluated associations between daytime napping / sleep duration and level of serum CRP among 5018 participants of the EPIC Norfolk study. The analysis is largely cross-sectional although the assessment for napping was a few years prior to CRP measurement. The authors found that day napping was associated with higher level of CRP while the association between nighttime sleeping/time-in-bed and CRP was u-shaped. This is the first study on napping and CRP, and its findings may help understand some of the previous epidemiological observations on daytime napping and adverse health outcomes among the elderly. Overall the analyses were well performed, particularly the joint analysis on daytime napping and nighttime sleeping. The reviewer however has several comments for authors to consider.

1) Abstract: It is somewhat misleading to call this a prospective study given the short time interval between exposure and outcome and the one-time measurement of outcome.

Response: We agree that this is essentially cross-sectional analysis and have changed the text accordingly.

2) Given the average age of the cohort and the stronger findings among elderly, the discussion may be strengthened by commenting on previous findings on napping/sleeping and degenerative diseases, such as dementia and Parkinson's disease (e.g. Gao et al. AJE 2011).

Response: Thank you for the insightful comments and we have now extended our discussion on this matter (page 18).

3) Given the complex relationship between napping/sleeping and various outcomes, it is not a surprise to see attenuation in the multivariate model which might actually have resulted in some over adjustment; the authors may want to acknowledge this possibility in the discussion.

Response: We agree that the association could have been underestimated as a result of over adjustment, and have acknowledged this in the discussion (page 16).

4) Were any other pro-inflammatory markers measured in addition to CRP? Any QC for CRP measurement, if so, please provide the data.

Response: Unfortunately we did not measure other pro-inflammatory markers that have appeared in the literature. However, we agree with the importance of examining other pro-inflammatory markers in relation to sleep, and have acknowledged the need for further studies in the discussion (page 18). QC information for CRP measurement has now been added (page 7).

5) Table 3 footnote: no explanation for “*...”

Response: Thank you for pointing this out. The explanation has been added.

6) Table 4: there was a borderline statistical interaction between napping and preexisting

diseases, but the Betas from these two groups were identical, please make sure numbers are accurate.

Response: Thank you for the careful observation. We have repeated the analysis and have observed the same results. However, as the reviewer correctly pointed out, the interaction term was only borderline significant. Therefore, we have interpreted the results cautiously.

Reviewer: 2

Reviewer Name Teresa Ward

Comments:

The study evaluates daytime napping, sleep duration, and serum C-reactive protein in 5,018 adults 48 to 92 years. Daytime napping, sleep duration, and time in bed (TIB) were obtained via self-report and CRP was measured one time via serum. Covariates –age, sex, BMI, physical activity, smoking, alcohol intake, pre-existing diseases and conditions (systolic BP, depression), and demographics were controlled for in the analysis. Compared to non-nappers, those who napped during the day had increased CRP levels, however these were not statistically different. This finding was more pronounced in older adults, women, and increased TIB. Sleep duration was not a significant predictor of CRP. These findings may reflect the fact that CRP was collected one time after napping data was obtained, higher proportion of women in the sample, and/or 62% of your sample reported no napping.

A few comments:

1. Page 6, line 12—“Habitual daytime napping as a lifestyle etc” authors mention different physiological effects compared to napping as a form of recovery sleep. Given the broad age range in your sample (48 to 92 years), further discussion about the changes in sleep from middle adulthood and older adulthood would strengthen this manuscript. Elderly often experience advanced sleep phase and daytime naps are common.

Response: We think the point raised by the reviewer is important, and have further discussed age-related changes in sleep in the introduction (page 6).

2. Long TIB was associated with increased CRP, and short TIB was not. Further discussion on this finding would strengthen the discussion. amplitude as an arousal marker, wavelet analysis of EEG, etc While power analysis is better than standard stages, it is a limited approach.

Response: Reviewer’s point is well taken and we have extended the discussion on the finding on TIB (page 19).